# Improving Assessment of Cognitive Impairment after Spinal Cord Injury: Methods to Reduce the Risk of Reporting False Positives

**DOI:** 10.3390/jcm12010068

**Published:** 2022-12-21

**Authors:** Danielle Sandalic, Yvonne Tran, Mohit Arora, James Middleton, Candice McBain, Daniel Myles, Ilaria Pozzato, Ashley Craig

**Affiliations:** 1John Walsh Centre Rehabilitation Research, The Kolling Institute, Royal North Shore Hospital, St Leonards, Sydney, NSW 2065, Australia; 2Faculty of Medicine and Health, Sydney Medical School, The University of Sydney, Sydney, NSW 2006, Australia; 3SCI Unit, Royal North Shore Hospital, St Leonards, Sydney, NSW 2065, Australia; 4Australian Institute of Health Innovation, Macquarie University, North Ryde, Sydney, NSW 2113, Australia; 5Spinal Outreach Service, Royal Rehab, Ryde, Sydney, NSW 2112, Australia

**Keywords:** mild cognitive impairment, spinal cord injury, normative assessment, premorbid intelligence

## Abstract

Adults with spinal cord injury (SCI) are reported to have heightened risk of cognitive impairment, notably mild cognitive impairment (MCI). Reports of increased risk of MCI are almost exclusively based on cross-sectional assessments of cognitive function using norm-referenced scores. Norm-referenced single-point assessments do not reflect cognitive decline at the individual level but rather represent between group differences in cognitive function. The practice of relying solely on norm-referenced assessment to study MCI after SCI is therefore problematic as it lends to potential misclassification of MCI. Premorbid intelligence estimates permit comparison of people’s actual versus expected cognitive function and thereby can be used to validate the presence of genuine cognitive decline. These are not utilized in the assessment of MCI after SCI. This study simulated data for 500,000 adults with SCI to compare norm-referenced and premorbid-intelligence methods of screening for MCI to examine the potential extent of MCI misclassification after SCI resulting from the overreliance on norm-referenced methods and exclusion of premorbid intelligence methods. One in five to one in 13 simulated adults with SCI were potentially misclassified as having MCI showing that measures of premorbid cognitive function must be included in assessment of cognitive function after SCI.

## 1. Introduction

Mild cognitive impairment (MCI) is a diagnosis given when (i) subjective complaints of changes in cognitive function are expressed, (ii) objective evidence of impairment in one or more cognitive domains exists in spite of preserved functional independence, and (iii) there is no dementia [1,2,3,4,5]. Studies since the 1980s have proposed that risk of MCI increases after spinal cord injury (SCI) [6]. Systemic disturbances after SCI including but not limited to autonomic, cardiovascular, central and peripheral inflammatory responses are thought to affect cognitive function negatively after SCI [7]. SCI is a severe neurological injury that requires substantial adjustment due to multiple secondary conditions associated with the injury, and it is suspected that MCI compromises adjustment [8].

However, most if not all studies claiming there is an increased risk of MCI after SCI have cited neuropsychological test-score differences between individuals with and without SCI as evidence. Such between-person comparisons are typically cross-sectional comparisons based on normative standards (i.e., norm-referenced tests) and cannot speak to the intra-individual cognitive decline necessary for a MCI diagnosis. Studies into associations between MCI and SCI are therefore seriously limited as they ignore valid objective evidence of impairment by relying on cross-sectional methods not fit to assess within-person cognitive decline [9].

Furthermore, tests used to assess and research cognitive function in individuals with SCI have been normed in non-SCI samples. The Neuropsychiatry Unit Cognitive Assessment Tool (NUCOG) is the single exception [10], although studies utilizing this test have based analyses on norms developed during the scale’s validation in place of available SCI-specific norms [11]. NUCOG was validated using three diagnostic groups (dementia, neurological disorders and psychiatric illness including depression and psychosis), and a ‘healthy’ control group [12], so it is questionable whether these norms apply to individuals with SCI. Recent investigation of the structural validity of the NUCOG showed that two of its five scales (language and memory) had no fit in a SCI sample, and one scale (executive functioning) had poor fit [13]. Other tests used frequently to assess cognition after SCI such as the Montreal Cognitive Assessment (MoCA) and the National Institute of Health Toolbox Cognition Battery (NIHTB-CB) seem similarly suboptimal for differentiating impaired and normal cognition in the context of SCI as they have been normed in non-SCI samples. The problem of making diagnoses of MCI based on between-person test comparisons is exacerbated by the use of norms derived from samples unrelated to SCI.

Ideally, baseline data from a person’s previous neuropsychological assessments would be available and incorporated into research designs and clinical practice, thereby enabling personalized assessment of scores across time to establish the absence or presence of clinically meaningful cognitive decline [14,15,16]. Realistically, baseline data is rarely available in the clinical setting making it necessary to employ alternative strategies [17], such as reliable estimates of premorbid functioning that can guide person-centered assessments. Tests of premorbid functioning (TOPF) serve this purpose, taking the form of lexical tasks, given that vocabulary is correlated highly with other cognitive functions and considered the best single measure of global intelligence [18,19]. TOPFs qualify as ‘hold’ tests as it is believed impairments such as traumatic brain injuries do not significantly compromise the reliability of their scores [20]. Arguably, TOPFs are indispensable in determining the presence of cognitive decline but are rarely utilized in the context of SCI, at least in the relevant published literature. The idea that TOPFs could help to improve the reliability of estimates of the extent of MCI after SCI requires urgent research attention.

Two assessment strategies can be used to assess cognitive function after SCI: (i) the normative mean method (nM-method) which compares individuals’ test scores against the mean of a normative group, and (ii) the premorbid intelligence method (pIQ-method) which compares individuals’ test scores against their estimated premorbid intelligence. This study performs comparisons between these two methods, primarily to estimate percentages of disagreement that can occur when the nM-method is used at the exclusion of the pIQ-method (i.e., the nM-method identifies a score as being impaired when the pIQ-method does not and vice versa). Such disagreements reflect potential MCI misclassifications.

By applying simulation procedures developed by Gavett et al. [21], this study simulated the possible extent of nM-/pIQ-method disagreements across the reported global prevalence of SCI. It was hypothesized that varying the standard deviation required for the identification of MCI (1, 1.5 and 2 SDs) and the base rate of estimated MCI (0.1, 0.3, and 0.6) would affect rates of disagreement between the two assessment methods. The application of different criteria achieved the purpose of identifying methodological and modifiable sources of variability in reported prevalence of MCI after SCI, which currently ranges from one in ten to six in ten adults with SCI. This study intended to highlight the problem of heterogeneity between studies of cognitive function after SCI to appeal for standardization of assessment practices.

## 2. Materials and Methods

Baseline data of 62 adults participating in an inception cohort longitudinal study of cognitive function after SCI informed all simulation procedures. Longitudinal study data (see Table 1) included socio-demographic factors (age, sex, years of education), and scores on the Neuropsychiatry Unit Cognitive Assessment Tool (NUCOG) and the Test of Premorbid Functioning. The power of the longitudinal study was calculated at 84%. This was based on size of the sample, an α of 0.05, and an estimated small to moderate effect size of size of 0.3.

The longitudinal study mean NUCOG value of 91.74 (see Table 1; rounded to 92) was used to simulate scores for 500,000 adults with SCI, as this is the reported global prevalence of SCI [22], and the mean of a large sample moves closer to the mean of the whole population. This mean NUCOG score was very close to the mean score reported in the NUCOG manual [12] and NUCOG SCI validation studies [10]. Simulation methods developed by Gavett et al. [21] were applied to the longitudinal study data to simulate the socio-demographics and NUCOG and Test of Premorbid Functioning scores for the 500,000 simulated adults with SCI, and these were used to compare the nM- and pIQ- methods of MCI assessment. Standard deviations (SDs) of 1, 1.5, and 2 were used to evaluate how different cut-offs for impairment applied under the nM-method affected the rate of nM-method/pIQ method disagreement. Under the pIQ-method, a NUCOG score was classified as impaired if it deviated from a person’s estimated premorbid level of cognitive function.

### 2.1. The Inception Cohort Longitudinal Study

Commenced in December 2019, the inception cohort longitudinal study followed adults with SCI from the first 24 to 48 h of their presentation to an emergency department (ED). Participants were assessed using cognitive and psychosocial measures across the ED and acute stages, through SCI rehabilitation and discharge from hospital, and up to 12-months post-injury. The assessments were performed in one of three specialized SCI rehabilitation units (i.e., Prince of Wales Hospital, Royal North Shore Hospital, and Royal Rehab) in New South Wales, Australia, and in the community at 12-months from being discharged from these units [23].

### 2.2. Longitudinal Study Participants

The longitudinal study participants were adults with acute SCI. They were recruited when engaged in SCI rehabilitation in one of the three above-mentioned SCI units. Inclusion criteria consisted of: (i) age 17–80 years; (ii) acute SCI of non-traumatic or traumatic origin, and (iii) sufficient proficiency in the English language to complete assessments. Exclusion criteria comprised (i) the presence of a severe mental disorder (e.g., bipolar disorder or schizophrenia) and (ii) the presence of a severe pre-morbid or concurrent brain injury (loss of consciousness > 24 h, post-traumatic amnesia > 7 days, or a Glasgow Coma score of 3–8 usually assessed within 24 h of initial injury).

### 2.3. Longitudinal Study Measures

Socio-demographic measures comprised age, sex, and years of education. Injury characteristics included level of injury and completeness of the lesion, assessed by a medical specialist based on the International Standards for Neurological Classification of SCI (http://ais.emsci.org/, accessed on 6 April 2020). Cognitive function was assessed by the NUCOG. The NUCOG is a validated cognitive screen comprising 21 items grouped into five cognitive domains: attention, perceptual/visuo-constructional, memory, executive and language. Scores for domains range from 0–20, with a combined total NUCOG score of 100, and with higher scores indicating higher levels of cognitive function.

Senior neuropsychologists were involved in the development of the NUCOG, and it is based on multiple neurocognitive tests such as the Stroop, Trail Making Test, and WAIS-4th Edition. The NUCOG has demonstrated criterion, convergent and discriminant validity between SCI and able-bodied samples, and acceptable reliability and specificity/sensitivity. When applied to screening adults with SCI for possible cognitive impairment, items needing normal hand function (e.g., drawing reproduction) require adaptation as per previous studies where this has been shown not to alter the validity of NUCOG scores [11]. When face-to-face administration of the NUCOG was not possible in the longitudinal study (e.g., discharge to the community outside metropolitan areas within the state of NSW, Australia, or social distancing restrictions due to COVID-19), administration occurred via telehealth methods. Administration of the NUCOG via teleconferencing required additional adjustments to the NUCOG assessment procedures to satisfy the telehealth environment. These procedural adjustments have been discussed elsewhere. Although the NUCOG has been shown to have poor structural fit in the Memory and Language domains in a SCI sample, the overall NUCOG score has demonstrated adequate validity [13].

The Test of Premorbid Functioning [24] was used to estimate pre-morbid cognitive functioning. This involved reading a list of 70 phonemically irregular words and was scored as per testing manual instructions. The numbers of words read aloud with a correct pronunciation (raw score) were transformed into age-corrected standard scores.

### 2.4. Simulation Parameters

As stated above, this study applied simulation procedures to the longitudinal study baseline data to generate socio-demographic factors and NUCOG and Test of Premorbid Functioning scores for a simulated sample of 500,000 adults with SCI. Hence, the simulated data reported hereafter is based on the socio-demographics and obtained NUCOG and Test of Premorbid Functioning scores of the original longitudinal study sample of 62 adults with SCI. The socio-demographics and NUCOG and Test of Premorbid Functioning scores for the longitudinal study sample are presented in Table 1.

Simulation parameters for the number of standard deviations required to classify a NUCOG score as ‘impaired’ (1 SD, 1.5 SD and 2 SD) and the assumed base rate of MCI in the global SCI population (0.1, 0.3 and 0.6) were theoretically guided. According to Diagnostic and Statistical Manual of Mental Disorders Fifth Edition (DSM-5) criteria, 1–2 SDs from ‘normal’ is the range of scores required for a diagnosis of mild neurocognitive disorder. Worldwide, 10–60% of adults are reported to display cognitive impairment after SCI [6,9].

### 2.5. Simulation Procedures

Simulations were conducted in R version 4.0.4 (R Foundation for Statistical Computing, Vienna, Austria) (https://www.npackd.org/p/r/4.0.4, accessed on 31 October 2022) by adapting the assumptions and code available in Gavett et al. Our assumptions are listed as follows, while those of Gavett et al. are presented in brackets for ease of reference. Where there was no difference in assumptions, the words ‘no difference’ indicates this: (1) the population mean premorbid IQ corresponding to a z-score of 0 was 100 (no difference), (2) the population mean cognitive test score was 91.7, rounded to 92, (the population cognitive test score was 89.5), (3) the correlation between the NUCOG cognitive test and premorbid IQ was 0.42 (similar to Gavett), (4) the effect size was 0.7 (no difference), (5) there was an average of 13 years of education (no difference), (6) the internal reliability of the cognitive test was 0.90 (no difference).

Simulated scores were obtained to identify percentages of nM-/pIQ-method disagreements. The cut-scores for classifications of “impaired” were manipulated by varying the standard deviations below the expected scores and the base rates of MCI in the SCI population (0.1 = low, 0.3 = medium and 0.6 = high). These base rates were chosen to reflect the highly heterogeneous rates of MCI reported in studies of cognitive function after SCI. Therefore, there were three possible scenarios affecting classifications of cognitive function as “impaired” or “normal” for both the NUCOG and Test of Premorbid Functioning based on simulated observations. The number of true positives and the disagreement rates were generated. For in-depth detail regarding the statistical and simulation calculations, applied readers are referred to Gavett et al. [21].

## 3. Results

Data regarding the percentages of disagreement between the nM-method and the pIQ-method of screening for potential MCI diagnoses using the NUCOG and Test of Premorbid Functioning are shown in Table 2. These data reveal that disagreements between the two methods of assessment (under the simulated parameters) ranged from 7.7% to 23.4%. The rate of disagreement between the two methods of assessment depended on the chosen SD cut-off used to classify a NUCOG score as impaired and the assumed base rate MCI in the SCI population.

As can be seen, increases in the base rate of MCI increased the likelihood of a disagreement between the nM-method and the pIQ-method of MCI assessment when the cut-off to classify MCI was more conservative (1.5 SD and 2 SD). When the cut-off to classify MCI was broad (1 SD), a trend was found in which increases in the base rate of MCI resulted in reduced likelihood of a disagreement between the nM- and pIQ-methods.

## 4. Conclusions

This study compared the nM- and pIQ-methods of MCI assessment in a simulated sample of adults with SCI, assuming that exclusive reliance on the nM-method increases the risk of false positive MCI classification on account of failing to consider a person’s premorbid cognitive function. The study supported the assumption. Worryingly, the exclusive use of the nM-method resulted in one in five (1:5) to one in 13 (1:13) potential false positive MCI classifications, and this was evident only with the inclusion of the pIQ-method.

Results showed that the base rate of MCI and number of SD used to classify impairment affected the rate of disagreement between the nM- and pIQ-methods. Specifically, the likelihood of disagreement between the two methods was highest when the SD used to classify impairment was smaller (1 SD), and this was largely irrespective of changes in base rate. Selecting one SD below the normative mean to classify MCI after SCI appears unsuitable given that it appeared most susceptible to identifying false positive MCI cases. When the SD used to classify impairment was conservative (i.e., 1.5–2 SD), the likelihood of disagreement between nM-method and pIQ-method was reduced, although increasing base rates increased the likelihood of disagreement between methods.

These findings support the use of a comprehensive method to diagnose MCI after SCI that includes norm-referenced and premorbid cognitive function measures, or at least reasonable proxies of premorbid cognitive function such as neuropsychological test histories that can verify the presence of actual cognitive decline. The current trend of investigating cognitive function after SCI without reference to markers of baseline cognitive status has been shown to introduce false positives in reported rates of MCI after SCI and this requires rectification. Research to compare different assessment criteria and practices is required to identify a set of criteria that is least conducive to false positive MCI diagnoses. This is the first study to have empirically investigated the influence of differing assessment criteria on the identification of MCI after SCI. Studies are needed to develop clinical and neuropsychological testing practices that optimize the balance between sensitivity and specificity when diagnosing MCI in the context of SCI.

It must be noted that several factors or study limitations could have possibly influenced the rate of disagreement found between the two assessment methods. Firstly, it must be noted that the NUCOG is not a gold standard screen for MCI assessment and therefore it cannot be considered a benchmark that fully reflects the merits of the nM-method of MCI assessment. Secondly, the correlation between the NUCOG and the Test of Premorbid Functioning in this simulated sample of adults with SCI was 0.42, showing that the cognitive screen and premorbid measure tapped related but not equivalent constructs, and this would have increased the likelihood of disagreement between the nM- and pIQ-methods.

The NUCOG has been shown to have suboptimal structural validity in an SCI sample, with language and memory domains having no fit and the executive functioning domain having poor fit [13]. Therefore, the differences between the two assessment methods might be specific to use of the NUCOG. This possibility should be investigated by future researchers who could repeat this study using a wider selection of neuropsychological tests to compare the nM- and pIQ- methods of assessment.

In their simulation study, Gavett et al. [21] reported that a lower correlation of 0.22 between a screen of cognitive function and test of premorbid intelligence favored the nM-method, whereas moderate to large correlations of 0.44 and 0.66 favored the pIQ-method. Our opinion proffered here is that in the absence of longitudinal cognitive function data, each approach is necessary to avoid misclassifying people with historically poor cognitive function or failing to identify cognitive decline in people with high premorbid cognitive function who can score in the normal range despite genuine cognitive decline. There appears to be no practical reason to favor either approach to MCI assessment given the relative ease of including a test of premorbid cognitive function which generally takes five minutes to administer and two to three minutes to score.

The importance of including tests of premorbid cognitive function in the assessment of MCI is illustrated best by the results of the Scottish Mental Surveys (SMS) of 1932 and 1947, which demonstrated the stability of premorbid cognitive function as a predictor of cognitive function in later life. These studies compared the intelligence scores obtained by two large cohorts at age 11 years with the intelligence scores they obtained over 60 years later. They found that childhood intelligence accounted for approximately 50% of intelligence at age 70 years [25,26]. Thus, neglecting to assess premorbid cognitive function potentially risks mistaking almost half of the variability of an obtained cognitive test score for some other variable(s) at any point in time.

Although premorbid cognitive function predicts a significant percentage of a person’s cognitive function at any age, higher premorbid cognitive function does not necessarily protect against cognitive decline. The SMS failed to find a relationship between childhood intelligence and rates of cognitive decline, and other studies have echoed these findings. Tucker-Drob and Salthouse [27] attributed approximately 40% of individual differences in longitudinal cognitive change to a domain-general factor which could represent premorbid global intelligence, and 30% each to domain-specific factors and variation in cognitive tests. This suggests that people’s prior experiences or exposure to a skill or practice affect their performance on cognitive screens, and it should be kept in mind that some experiences or skills may provide an advantage or disadvantage to performance. There is a need to consider MCI assessment from a life course perspective that accounts for the multiple influences on cognitive function and not merely from the perspective of any single cross-sectional assessment method, especially where this relies solely on normative references.

Finally, the results of this study provide evidence to tighten controls for reporting and use of the label ‘cognitive impairment’, particularly when results represent cross-sectionally obtained scores and there is no way of confirming the presence of genuine cognitive impairment. The reporting of ‘impairment’ should be reserved for confirmed MCI cases, arguably to avoid misrepresenting the extent of MCI after SCI. Presently, the range of MCI after SCI is reported to be between 10–60% [6]. This range is far too broad to be clinically meaningful or reflect the scope of the problem of MCI after SCI. When results of cognitive screens at a cross-section of time are reported, it may be better to report scores more objectively with statements restricted to the observed data (e.g., adults with SCI score lower on the MoCA or NUCOG on average than do adults without SCI).

In conclusion, this is the first study to have examined the potential implications of relying on the nM-method of MCI assessment at the exclusion of the pIQ-method after SCI. The disadvantages of sole reliance on the nM-method of MCI assessment demonstrated in this study are unlikely to be limited to use of the NUCOG. There is an urgent need to examine how the nM- and pIQ-methods compare when other frequently used cognitive screens such as the MOCA are applied to the assessment of MCI after SCI. To strengthen the validity of norm-referenced MCI assessment, it will be necessary to develop SCI norms for existing neuropsychological screens, and to devise new tests that are sensitive to the unique testing needs of this population.

## Figures and Tables

**Table 1 jcm-12-00068-t001:** Socio-demographic, injury characteristics, NUCOG total, NUCOG domains, and Test of Premorbid Functioning raw and standard scores for the 62 SCI participants.

	Mean (SD)	±95% CI	Frequency
Age (years)	51.27 (18.4)	46–56	--
Sex			
Male			49 (79)
Female			13 (21)
Years of Education *	13.44 (2.8)		--
Level of Injury			
Cervical			33 (53.2)
Thoracic			24 (38.7)
Lumbar/Sacral			5 (8.1)
ASIA			
A			14 (22.6)
B			9 (15.5)
C			21 (33.9)
C			18 (29.0)
NUCOG Total	91.74 (5.8)	90–93	--
NUCOG Attention	17.39 (2.4)	17–18	--
NUCOG Memory	17.79 (1.9)	17–18	--
NUCOG Executive	17.71 (2.6)	17–18	--
NUCOG Perceptual	19.30 (1.2)	19–20	--
NUCOG Language	19.54 (0.7)	42–50	--
TOPF Raw Score	46.26 (14.7)	42–50	--
TOPF Standard Score	104.53 (14.7)	101–108	--

* indicates there are 17 missing values for Years of Education; ASIA: American Spinal Injury Association International Standards for Neurological Classification of SCI.

**Table 2 jcm-12-00068-t002:** Rates of disagreement between the the nM- and pIQ-methods of MCI Assessment.

	10% Base Rate	30% Base Rate	60% Base Rate
1 SD	23.4%	23.1%	22.9%
1.5 SD	14.8%	16.2%	18.3%
2 SD	7.7%	9.6%	12.2%

Note. These rates are based on a correlation of 0.42 between the NUCOG and Test of Premorbid Functioning, a mean NUCOG score of 92 (rounded up from 91.74), an SD of 5.8, and an internal consistency of at least 0.9 for the NUCOG.

## Data Availability

The datasets generated are available from the corresponding author on reasonable request.

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
