# Peer review of "Improving Assessment of Cognitive Impairment after Spinal Cord Injury: Methods to Reduce the Risk of Reporting False Positives"

_jcm, 2022, doi:10.3390/jcm12010068_

Round 1
Reviewer 1 Report
This study is edifying, well-written, and thoroughly researched in comprehensive in order to reduce the false positive reporting of cognitive impairment in spinal cord injured patients.
The manuscript “Improving assessment of cognitive impairment after spinal 2 cord injury: Methods to reduce the risk of reporting false positives” the authors investigated the advantages of using premorbid intelligence method along with normative mean method to reduce the false positive reporting of cognitive impairment in Spinal Cord Injured patients.
I have a few minor comments for the authors to improve the manuscript
1. Typos
a. Font type/size difference - Page 4 – Line 156 Test of Premorbid functioning
b. Font type/size difference - Page 4 – Table 1
c. Page 4 – Table 1 – space between ±95% CI
d. Page 4 – Table 1 - Specify the unit (Age in Years)
e. Font type/size difference - Page 4 – Line 156 – Test of Premorbid functioning (Age in Years)
f. Font type/size difference - Page 5 – American Spinal Injury Association Internal Standards for Neurological Classification of SCI
g. Page 5 – Line 198 *17 – not clear (rewrite clearly)
h. Page 201 – 1SD, 1.5SD, and 2SD and in Table 2 written as 1 SD, 1.5 SD, and 2 SD – Make it uniform (either one format).
i. The term "MOCA" on Page 2 Line 62 and page 7 Line 312, the word is abbreviated in different forms. Make it uniform.
2. Introduction
a. Page 2- Line 74 “Tests of ……. intelligence”
3. Methods
a. Page 2 Line 96, Specify the criteria/formula to calculate the sample size in order to justify the required minimum sample size.
b. Page 3, Line 135 Rather than mentioning the URL, why not include it in the reference?
c. Page 5, Line 208 Rather than mentioning the URL, why not include it in the reference?
d. Page 5, Line 212, Scatter graph/diagram can be included for the correlation.
4. Discussion
a. Page 6 – Line 261, I have no substantial remarks about the gold standard of NUCOG. To support this statement, I recommended adding some clear pertinent literature to the statement and ideally discussing the suboptimal structural validity (language and memory and executive functioning domain) to enhance the value of this manuscript..
Author Response
Response to Reviewer 1 Comments
Dear Reviewer,
Thank you for reviewing our manuscript and your helpful suggestions for revision.
Please find our reply to your comments.
- Typos
- Font type/size difference - Page 4 – Line 156 Test of Premorbid Functioning
Response 1a: I have reviewed the Word version of the manuscript at the section that corresponds to Page 4 – Line 156 and there is no font/size difference on the Word file. The reviewer is correct as a size difference does appear on the converted PDF version. This distortion seems to have occurred during the conversion process and I am not sure I am able to address this.
- Font type/size difference - Page 4 – Table 1
Response 1b: The table was not formatted as a Table and this appeared to contribute to the identified problem. The correct formatting has now been entered and should address this problem.
- Page 4 – Table 1 – space between ±95% CI
Response 1c: I have entered a space between ±95 and CI.
- Page 4 – Table 1 - Specify the unit (Age in Years)
Response 1d: The table now reads “Age (years).
- Font type/size difference - Page 4 – Line 156 – Test of Premorbid Functioning (Age in Years)
Response 1e: This comment is the same as comment 1a. Please see response to 1a.
- Font type/size difference - Page 5 – American Spinal Injury Association Internal Standards for Neurological Classification of SCI
Response 1f: I have reviewed the Word version of the manuscript at the section that corresponds to Page 5 – Line 198 and there is no font/size difference on the Word file. The reviewer is correct as a size difference does appear on the converted PDF version. This distortion seems to have occurred during the conversion process and I am not sure I am able to address this.
- Page 5 – Line 198 *17 – not clear (rewrite clearly)
Response 1g: Line 198 now reads “*indicates there are 17 missing values for Years of Education; ASIA: American Spinal Injury Association International Standards for Neurological Classification of SCI.” This change hopefully makes line 198 read more clearly.
- Page 201 – 1SD, 1.5SD, and 2SD and in Table 2 written as 1 SD, 1.5 SD, and 2 SD – Make it uniform (either one format).
Response 1h: Spaces have been inserted at line 201 so it reads 1 SD, 1.5 SD, and 2 SD to make it uniform with presentation in Table 2.
- The term "MOCA" on Page 2 Line 62 and page 7 Line 312, the word is abbreviated in different forms. Make it uniform.
Response 1i: The abbreviation of the Montreal Cognitive Assessment has been changed to MoCA at Line 312 so that this abbreviation matches the abbreviation that appears at Line 62.
- Introduction
- Page 2- Line 74 “Tests of ……. intelligence”
Response: Line 74 of the manuscript reads Tests of premorbid functioning (TOPF) serve this purpose. Please excuse me as I do not understand what the reviewer means by his comment “tests of …….. intelligence”.
- Methods
- Page 2 Line 96, Specify the criteria/formula to calculate the sample size in order to justify the required minimum sample size.
Response 2a: Thank you for this suggestion. Line 96 onwards now reads:
Baseline data of 62 adults participating in an inception cohort longitudinal study of cognitive function after SCI informed all simulation procedures. Longitudinal study data (see Table 1) included socio-demographic factors (age, sex, years of education), and scores on the Neuropsychiatry Unit Cognitive Assessment Tool (NUCOG) and the Test of Premorbid Functioning. The power of the longitudinal study was calculated at 84%. This was based on size of the sample, an α of 0.05, and an estimated small to moderate effect size of size of 0.3.
- Page 3, Line 135 Rather than mentioning the URL, why not include it in the reference?
Response 2b: Thank you for this suggestion. We have left the URL in place for ease of reference.
- Page 5, Line 208 Rather than mentioning the URL, why not include it in the reference?
Response 2c: Thank you for this suggestion. We have left the URL in place for ease of reference.
- Page 5, Line 212, Scatter graph/diagram can be included for the correlation.
Response 2d: Thank you for the suggestion. We believe a scatter diagram is not required to illustrate this simple correlation.
- Discussion
- Page 6 – Line 261, I have no substantial remarks about the gold standard of NUCOG. To support this statement, I recommended adding some clear pertinent literature to the statement and ideally discussing the suboptimal structural validity (language and memory and executive functioning domain) to enhance the value of this manuscript.
Response 4a: Thank you for this comment. We have stated that the NUCOG is not a gold standard test and reported this as a potential study limitation as the differences found between the normative and premorbid methods of assessing cognitive impariment may have reflected use of the NUCOG rather than the normative method per se. Using a different neurocognitive test may have revealed no or other sources of difference between the normative and premorbid methods. We have cited the only study that has examined the structural validity of the NUCOG, which is one of our former studies appearing as Reference 13 at Line 265 and believe this addresses the issues raised by Reviewer 1.
This said, we have also edited the discussion to address your comment. The following text appears betweeen Lines 255 to 270 of the original manuscript file:
It must be noted that several factors or study limitations could have possibly influenced the rate of disagreement found between the two assessment methods. Firstly, it must be noted that the NUCOG is not a gold standard screen for MCI assessment and therefore it cannot be considered a benchmark that fully reflects the merits of the nM-method of MCI assessment. Secondly, the correlation between the NUCOG and the Test of Premorbid Functioning in this simulated sample of adults with SCI was 0.42, showing that the cognitive screen and premorbid measure tapped related but not equivalent constructs, and this would have increased the likelihood of disagreement between the nM- and pIQ-methods.
The NUCOG has been shown to have suboptimal structural validity in an SCI sample, with language and memory domains having no fit and the executive functioning domain having poor fit(13). Therefore, the differences between the two assessment methods might be specific to use of the NUCOG. This possibility should be investigated by future researchers who could repeat this study using a wider selection of neuropsychological tests to compare the nM- and pIQ- methods of assessment.

Reviewer 2 Report
Dear authors, thank you for allowing me to revise your manuscript comparing the n-M and PIQ methods in assessing cognitive function after SCI. I found this manuscript well-written, engaging, and useful to guide future research and clinical practice.
Nevertheless, I raise some points that could improve the quality and usefulness of your study.
- In the introduction, please provide a clear rationale on why we should clearly know the best test to assess cognitive function. Please, provide a clear aim of the study.
- In the methods, and precisely in the simulation procedures section, you suggested reading Gavett et al., but I would (as an interested reader) know a little bit more on how the simulation was run. Moreover, you stated that "Base rates of MCI in the SCI population were also manipulated, with 0.1 as low, 0.3 as medium, and 0.6 as high base rates for MCI." what does it means from a research integrity perspective?
- In the discussion, I would expect to find some consideration about the applicability of the screening methods, as the implementation in the real setting of a method would be informed by real data. Moreover, I found that some information on the conduction of future studies in this field, considering your results, is lacking.
Author Response
Response to Reviewer 2 Comments
Dear Reviewer,
Thank you for reviewing our manuscript and your helpful suggestions for revision.
Please find our reply to your comments.
- In the introduction, please provide a clear rationale on why we should clearly know the best test to assess cognitive function. Please, provide a clear aim of the study.
Response 1: We have split the last paragraph of the introduction at Line 90 to include your request for a clearer statement of the study aims so the last paragraph now reads:
By applying simulation procedures developed by Gavett et al.(21) this study simulated the possible extent of nM-/pIQ-method disagreements across the reported global prevalence of SCI. It was hypothesized that varying the standard deviation required for the identification of MCI (1, 1.5 and 2 SDs) and the base rate of estimated MCI (0.1, 0.3, and 0.6) would affect rates of disagreement between the two assessment methods. The application of different criteria achieved the purpose of identifying methodological and modifiable sources of variability in reported prevalence of MCI after SCI which currently ranges from one in ten to six in ten adults with SCI. This study intended to highlight the problem of heterogeneity between studies of cognitive function after SCI to appeal for standardisation of assessment practices.
- In the methods, and precisely in the simulation procedures section, you suggested reading Gavett et al., but I would (as an interested reader) know a little bit more on how the simulation was run. Moreover, you stated that "Base rates of MCI in the SCI population were also manipulated, with 0.1 as low, 0.3 as medium, and 0.6 as high base rates for MCI." what does it means from a research integrity perspective?
Response 2: Please see the following changes to the section titled ‘Simulation Procedures’ for our response to your request.
Simulation Procedures:
Simulations were conducted in R version 4.0.4 by adapting the assumptions and code available in Gavett et al. Our assumptions are listed as follows while those of Gavett et al. are presented in brackets for ease of reference. Where there was no difference in assumptions, the words ‘no difference’ indicates this: 1) the population mean premorbid IQ corresponding to a z-score of 0 was 100 (no difference), 2) the population mean cognitive test score was 91.7, rounded to 92, (the population cognitive test score was 89.5), 3) the correlation between the NUCOG cognitive test and premorbid IQ was 0.42 (similar to Gavett), 4) the effect size was 0.7 (no difference), 5) there was an average of 13 years of education (no difference), 6) the internal reliability of the cognitive test was 0.90 (no difference).
Simulated scores were obtained to identify percentages of nM-/pIQ-method disagreements. The cut-scores for classifications of “impaired” were manipulated by varying the standard deviations below the expected scores and the base rates of MCI in the SCI population (0.1 = low, 0.3 = medium, and 0.6 = high). These base rates were chosen to reflect the highly heterogeneous rates of MCI reported in studies of cognitive function after SCI. Therefore, there were three possible scenarios affecting classifications of cognitive function as “impaired” or “normal” for both the NUCOG and Test of Premorbid Functioning based on simulated observations. The number of true positives and the disagreement rates were generated. For in-depth detail regarding the statistical and simulation calculations applied readers are referred to Gavett et al.(21)
- In the discussion, I would expect to find some consideration about the applicability of the screening methods, as the implementation in the real setting of a method would be informed by real data. Moreover, I found that some information on the conduction of future studies in this field, considering your results, is lacking.
Response 3: The following text has been inserted between what were Lines 255 to 270 in the original manuscript file. We hope this is an adequate response to your suggestion.
These findings support the use of a comprehensive method to diagnose MCI after SCI that includes norm-referenced and premorbid cognitive function measures, or at least reasonable proxies of premorbid cognitive function such as neuropsychological test histories that can verify the presence of actual cognitive decline. The current trend of investigating cognitive function after SCI without reference to markers of baseline cognitive status has been shown to introduce false positives in reported rates of MCI after SCI and this requires rectification. Research to compare different assessment criteria and practices is required to identify a set of criteria that is least conducive to false positive MCI diagnoses. This is the first study to have empirically investigated the influence of differing assessment criteria on the identification of MCI after SCI. Studies are needed to develop clinical and neuropsychological testing practices that optimise the balance between sensitivity and specific when diagnosing MCI in the context of SCI.
It must be noted that several factors or study limitations could have possibly influenced the rate of disagreement found between the two assessment methods. Firstly, it must be noted that the NUCOG is not a gold standard screen for MCI assessment and therefore it cannot be considered a benchmark that fully reflects the merits of the nM-method of MCI assessment. Secondly, the correlation between the NUCOG and the Test of Premorbid Functioning in this simulated sample of adults with SCI was 0.42, showing that the cognitive screen and premorbid measure tapped related but not equivalent constructs, and this would have increased the likelihood of disagreement between the nM- and pIQ-methods.
The NUCOG has been shown to have suboptimal structural validity in an SCI sample, with language and memory domains having no fit and the executive functioning domain having poor fit(13). Therefore, the differences between the two assessment methods might be specific to use of the NUCOG. This possibility should be investigated by future researchers who could repeat this study using a wider selection of neuropsychological tests to compare the nM- and pIQ- methods of assessment.

Round 2
Reviewer 2 Report
Dear authors, thank you for allowing me to review this second version of your manuscript. You provided very satisfactory answers to my comments. I am happy to recommend this manuscript for publication, as it is improved in its overall quality.